# WHICH LLMs GET THE JOKE? PROBING NON-STEM REASONING ABILITIES WITH HUMORBENCH

## ABSTRACT

We present HumorBench, a benchmark designed to evaluate large language models' (LLMs) ability to reason about and explain sophisticated humor in cartoon captions drawn from the New Yorker Caption Contest and Cartoonstock.com. As reasoning models increasingly saturate existing benchmarks in math and science, novel and challenging evaluations of model intelligence beyond STEM domains are essential. Reasoning is fundamentally involved in text-based humor comprehension, requiring the identification of connections between concepts in cartoons/captions and external cultural references, wordplays, and other mechanisms. HumorBench includes 300 unique cartoon-caption pairs from the New Yorker Caption Contest and Cartoonstock.com, with expert-annotated evaluation rubrics identifying essential joke elements. LLMs are evaluated based on their explanations towards the humor and abilities in identifying the joke elements. To perform well on this task, models must identify associations between concepts, potentially backtracking from initial interpretations to arrive at the most plausible explanation. Our extensive benchmarking of current models reveals three key insights: (1) LLM progress on STEM reasoning transfers effectively to humor comprehension; (2) models trained exclusively on STEM reasoning data still perform well on HumorBench, demonstrating strong transferability of reasoning abilities; and (3) scaling thinking token budgets yields mixed results across models in humor reasoning.

## 1 INTRODUCTION

Recent advances in large language models and reasoning techniques have led to the saturation of many existing benchmarks, particularly in STEM domains such as mathematics and programming, where frontier models now approach or exceed human-level performance (Abdin et al., 2025a; Sun et al., 2025; Quan et al., 2025). This progression highlights the need for novel and challenging evaluations that can meaningfully differentiate model capabilities and provide insights into their reasoning processes. Non-STEM reasoning tasks, particularly those involving cultural understanding and implicit knowledge, represent underexplored territories for model evaluation.

Humor comprehension represents a particularly challenging frontier for artificial intelligence (Hessel et al., 2023; Zhang et al., 2024; Zhou et al., 2025; Kazemi et al., 2025; Liang et al., 2025). Although large language models (LLMs) excel across many domains, understanding humor still requires sophisticated reasoning that integrates context, cultural knowledge, and implicit connections. These challenges make humor an ideal testbed for evaluating advanced reasoning in AI systems.

We present **HumorBench**, a benchmark that evaluates LLMs' ability to explain sophisticated cartoon-caption humor by identifying the mental leaps connecting visuals, captions, and external knowledge (Figure 1). For each pair, we annotate the objective elements essential for comprehension, creating a ground truth focused on factual connections rather than subjective appreciation.

We benchmark both a standard set and a harder subset. On HumorBench-hard, which features more complex examples requiring multiple reasoning steps or obscure cultural knowledge, no current LLM exceeds 60% accuracy. Our benchmarking of current state-of-the-art models reveals two key findings:

1. We observe a high correlation between performance on HumorBench and existing STEM benchmarks, suggesting a significant transfer of general reasoning abilities to humor comprehension tasks.

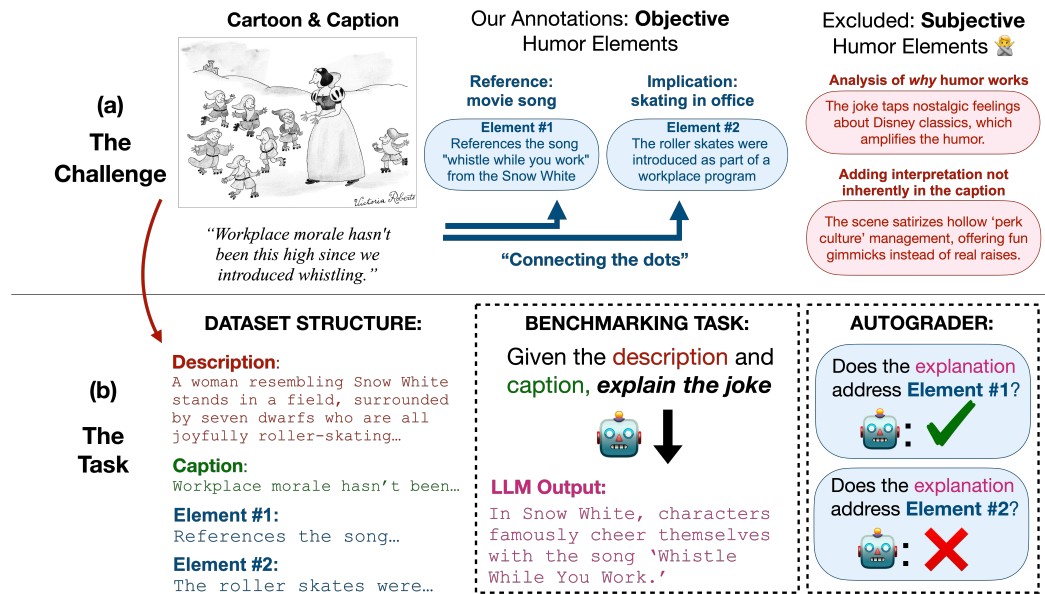

Figure 1: Overview of our humor analysis approach. (a) We distinguish between **objective** and **subjective** components of a joke. To convert the open-ended task of humor explanation into a fair benchmark, we focus exclusively on **objective** elements. (b) Overview of the dataset, benchmark task, and grading scheme in HumorBench. Each cartoon-caption pair contains one or more "element" annotation. For the benchmark, an LLM is tasked with explaining the joke in the caption. An autograder evaluates if the explanation contains each element.

2. Even models trained exclusively on STEM reasoning tasks (e.g., mathematical problem-solving) perform well on HumorBench, indicating that abstract reasoning skills acquired in one domain can transfer effectively to humor comprehension.

3. Test-time scaling measures for humor reasoning yield mixed results, indicating that simply increasing computational resources at inference time does not consistently improve performance on this challenging domain.

## 1.1 WHY ANOTHER LLM HUMOR BENCHMARK?

Several benchmarks have already focused on measuring LLMs' capabilities around humor. Specifically, Hessel et al. (2023) and Zhang et al. (2024) both build upon the New Yorker Caption Contest (NYCC) dataset—a weekly feature by the New Yorker magazine where readers submit funny captions for cartoons (see Figure 1 for an example). Hessel et al. (2023) created three benchmarks using this dataset: ranking the funniness of caption pairs, matching cartoons to valid captions, and explaining the humor behind captions. However, these benchmarks simultaneously measure two distinct capabilities: (1) understanding the intended jokes (objective elements) and (2) aligning with individual and subgroup humor preferences (subjective factors). As Zhou et al. (2025) points out, performance on these previous benchmarks is heavily influenced by an LLM's ability to align with specific audience preferences rather than directly measuring its reasoning about the jokes themselves.

Our benchmark, HumorBench, addresses this limitation by focusing solely on the objective elements of humor comprehension, specifically measuring the *humor reasoning abilities* required to understand cartoons and their captions. As validated by our experimental findings, LLMs' performance on HumorBench correlates well with their performance on other reasoning benchmarks.

We acknowledge HumorBench covers a somewhat narrow domain of humor: English-language single-panel cartoons from the New Yorker Caption Contest and Cartoonstock.com, which reflect a particular Western, New Yorker–style sense of humor. However, we believe this format offers a clean, well-controlled testbed for probing non-STEM reasoning in LLMs.

## 2 RELATED WORK

**Reasoning-focused language models.** Recent advances in large language models (LLMs) have seen the emergence of specialized reasoning models that excel at logical deduction, mathematical problem-solving, and multi-step reasoning while maintaining strong general language capabilities. These reasoning-enhanced models employ various approaches: training-focused methods like those used by *Minerva* (Lewkowycz et al., 2022), *WizardMath* (Luo et al., 2023), and *Phi-4-Reasoning* (Abdin et al., 2025b) leverage carefully curated STEM-heavy corpora; inference-time techniques boost reasoning without changing model weights, including *Self-consistency* (Wang et al., 2023), *Tree-of-Thought* methods in *DeepSeek-Math* (Shao et al., 2024), *Least-to-most* prompting (Zhou et al., 2023), and *Process supervision* (Lightman et al., 2023); while hybrid approaches like *MAmmoTH* (Yue et al., 2023) combine diverse training data with structured inference protocols, *ToRA* (Gou et al., 2023) integrates formal verification systems, and *MathGLM* (Yang et al., 2024) combines symbolic computation with natural language reasoning. Models like *Gemini Ultra* (Google, 2024) and *Claude 3.5* (Anthropic, 2024) achieve strong reasoning through both architectural innovations and sophisticated training, suggesting that advances in machine reasoning now follow multiple complementary paths rather than relying solely on parameter count (Wei et al., 2024).

**Humour benchmarks.** Beyond simple joke generation, several resources now probe LLM humour competence. Hessel et al. (2023) introduces three New Yorker cartoon-caption subtasks that test multimodal humour understanding and explanation. For word-play, the *ExPUNations* corpus augments classic pun datasets with human-written explanations and funniness ratings (Sun et al., 2022), while Xu et al. (2024) systematically benchmarks pun recognition, explanation and creation. Complementing these datasets, (Ermakova et al., 2025) Lab provides reusable test collections for humour-aware information retrieval.

**Open-ended evaluation frameworks.** Automatic grading of creative, unconstrained outputs increasingly relies on the *LLM-as-Judge* paradigm. G-EVAL couples chain-of-thought GPT-4 judging with a form-filling rubric, achieving human-level reliability on summarisation and dialogue (Liu et al., 2023). MT-BENCH and its crowdsourced *Chatbot Arena* show that GPT-4 judges agree with human preferences on multi-turn instruction following in ∼80% of cases (Zheng et al., 2023). Going further, PAPERBENCH grades agents on reproducing ICML-level research papers with hierarchical GPT-4 rubrics and expert audits (Starace et al., 2025). We adopt a similar rubric-guided judging scheme but focus specifically on humour reasoning, enabling systematic comparison of explanation quality across models.

## 3 HUMORBENCH

### 3.1 MAIN BENCHMARK TASK

HumorBench frames humor understanding as an open-ended task: given a textual description of a cartoon and its caption, a model must articulate in its own words the underlying joke. We deliberately avoid the multiple-choice or ranking formats common in existing humor benchmarks because, for creative tasks, fixed answer sets can (i) inadvertently hint at the punchline and (ii) fail to accommodate the diverse range of valid explanations a competent reader might produce.

To make this free-form setting automatically gradable, we distill each cartoon into a concise rubric of 1–3 objective "elements." An element represents a single, easily verifiable fact that any correct explanation must include (e.g., in NYCC Contest #665, the observation that "the shark interprets the swimmer as groceries"), as shown in Figure 8. This approach allows for creative expression while maintaining consistent evaluation standards (see Appendix A for the complete prompt).

### 3.2 DATASET: CARTOON AND CAPTION SOURCES

Our dataset comprises cartoons and captions from two primary sources: the New Yorker Caption Contest (NYCC) and Cartoonstock.com. We sourced NYCC captions from publicly available datasets (Hessel et al., 2023; Jain et al., 2020; Zhang et al., 2024), selecting only those ranked among the top 3 finalists to ensure each cartoon features a coherent, high-quality joke. For Cartoonstock cartoons, we utilized their original accompanying captions. Both sources specialize in dry, witty humor

that demands sophisticated reasoning—often requiring multiple mental leaps to fully comprehend, as illustrated in Figure 1.

While cartoons inherently include visual elements, our benchmark focuses on testing humor comprehension rather than visual interpretation capabilities. Therefore, we created detailed textual descriptions of each cartoon, carefully capturing all information necessary to understand the caption while maintaining neutrality. These descriptions include essential details about the setting, characters, visible emotions, and speaker identification, while deliberately omitting artistic style unless directly relevant to the joke. The choice to not directly include the images makes HumorBench a cleaner source of signal for humor reasoning, and not vision capabilities. For researchers interested in extending this to a multimodal benchmark, we provide source links to the original images: NYCC images are available through (Hessel et al., 2023; Jain et al., 2020), while Cartoonstock images require licensing.

## 3.3 DATASET: ELEMENT ANNOTATION

The core labels in our dataset are the element annotations assigned to each cartoon–caption pair. For every pair, we hand-annotated one to three elements—concise, direct statements that capture the objective components essential to understanding the joke. As discussed in Section 1.1, comprehending cartoon humor requires two distinct capabilities: understanding the objective content of the joke and recognizing the subjective aspects that influence audience reception. Figure 1 illustrates this distinction—subjective explanations focus on audience reactions (which vary between individuals), while objective elements center on content comprehension. Operationally, we treat an element as objective if it states a concrete, checkable fact or relation needed to reconstruct the intended reading of the joke from the description and caption, and treat as subjective anything that appeals to taste, funniness, or a particular theory of why the joke works. Our benchmark specifically targets these objective elements, which require identifying the mental leaps necessary to "get" the joke through recognizing references, wordplay, implications, or similar mechanisms. The autograder then evaluates LLM explanations against these elements, verifying that each explanation adequately covers the fundamental objective components of the humor, ensuring a fair and consistent assessment.

In summary, to make this task easily gradable, annotations follow deliberate guidelines: (1) Elements must be short, direct, and verifiable from the description and caption; (2) An element addresses exactly one concept. Bundling ideas may add noise by forcing the grader to guess about partial correctness; (3) Elements deliberately avoid adding noise from subjective opinions about humor.

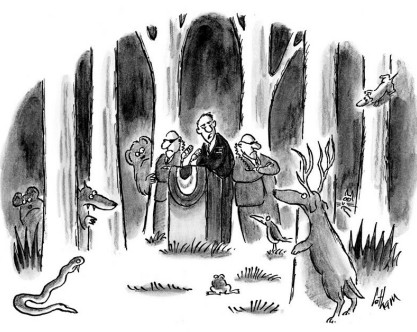

**NYCC Contest 167**
**Description:** In a forest clearing with tall tree trunks, a man in a suit with an American-flag lapel pin speaks at a podium while aides stand behind him. Woodland animals—a deer, snake, frog, and birds—peek from the trees and grass, watching.

**Caption:** "As a weasel, I need your vote."

---

**ELEMENT 1:** References the cliché insult of calling politicians *weasels*.

**ELEMENT 2:** Plays on the dual meaning of "weasel" (literal animal & political pejorative), creating a pun.

Figure 2: Example HUMORBENCH annotation. The cartoon (*left*) is paired with its description, caption, and two hand-labeled joke elements (*right*).

## 3.4 DATASET REFINEMENT

For an LLM evaluation to provide trustworthy results, the underlying dataset must be both accurate and internally consistent. We initially collected 655 unique element annotations, but despite careful guidelines, some entries proved vague or imprecise. To systematically improve quality, we implemented an iterative refinement process.

First, we generated sample explanations for each cartoon–caption pair, alternating randomly between GPT-4o and Claude 3.7 Sonnet. Each explanation was evaluated ten times by our autograder, with elements showing verdict disagreement exceeding 30% flagged for review. These problematic cases were either refined or removed entirely. We repeated this quality control cycle until fewer than 5% of annotations triggered inconsistency flags, ultimately resulting in 499 high-quality unique element annotations forming the foundation of HumorBench.

As an additional validation step, we invited a former chief cartoon editor of the New Yorker to review a random subset of 30 annotations. The editor confirmed that all elements were fair and accurately captured the essential components of each joke. Together, these atomic, objectively verifiable criteria create a robust rubric that enables our autograder to provide consistent and reliable evaluation at scale.

### 3.5 AUTOGRADER AND EVALUATION

During evaluation, an LLM judge assesses each model's explanation against individual elements to determine whether they adequately cover the essential components of the joke. This approach allows us to efficiently evaluate open-ended text generation at scale.

However, ensuring autograder consistency presents challenges, particularly for tasks that are inherently difficult for LLMs to understand (Min et al., 2020; Starace et al., 2025). To address this, we created a separate benchmark of 300 human expert judgments on explanations from three distinct LLMs: GPT-4o, Gemini 2.5 Pro, and Claude 3.7 Sonnet. Using GPT-4o as the autograder, we achieved 92% accuracy overall:

| Explainer Model | Acc. (%) | FPR (%) | FNR (%) |
|---|---|---|---|
| *Overall* (n=300) | 92.00 | 14.79 | 6.51 |
| Gemini 2.5 Pro | 93.00 | 10.00 | 6.25 |
| GPT-4o | 92.00 | 14.81 | 5.48 |
| Claude 3.7 Sonnet | 91.00 | 19.57 | 7.80 |

Table 1: Autograder performance (GPT-4o judge) on 300 human-labeled explanations.

This validation provides two key insights. First, across all models, the autograder's false positive rate (FPR) substantially exceeded its false negative rate (FNR), indicating a leniency bias. This suggests that *HumorBench scores should be interpreted as an upper bound on model performance*. Second, despite using GPT-4o as the autograder, we observed no significant advantage for GPT-4o-generated explanations compared to those from other models. Together, these findings confirm that our autograder provides a valid, albeit slightly optimistic, mechanism for large-scale evaluation.

**Length Control.** While models were instructed to keep responses under 200 words, some models exceeded this limit, particularly when reasoning traces were included in the final output. To ensure fair comparison, we truncated all model outputs to the last 1000 tokens.

## 4 EXPERIMENTS

Along with creating the HumorBench evaluation, we extensively benchmarked current frontier models. For consistency, all models are given the same prompt and scaffolding describing the task (see Appendix A). We arrived at this prompt after validating across several different LLMs (Claude 3.7 Sonnet Anthropic (2025), GPT-4o OpenAI (2024a), Gemini 2.5 Pro DeepMind (2025)). While many LLMs had different API endpoints, we tried to maintain consistent parameters where possible. For example, all models had temperature set to 1 and external tool calling deactivated. Note, for all evaluations, autograders, and benchmarks, "GPT-4o" refers to the gpt-4o-2024-08-06 release.

### 4.1 MAIN RESULTS

In general, the results from the main benchmarking effort were unsurprising. As shown in Figure 3, OpenAI o3 OpenAI (2025a) leads the pack at 87.5% accuracy, dramatically ahead of other SOTA

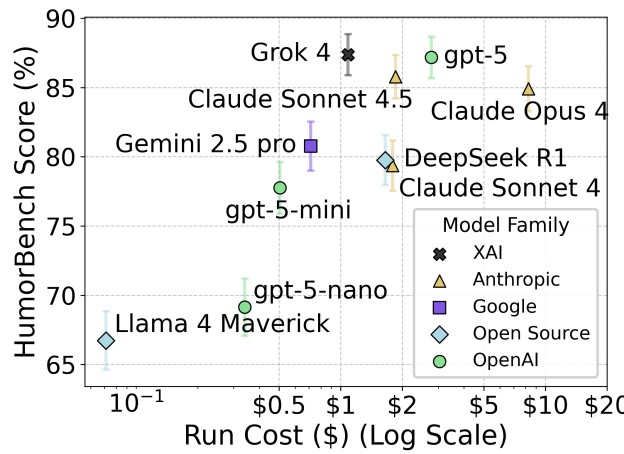

Figure 3: Benchmarking results on frontier models on HumorBench. Points show mean element-level accuracy; error bars denote ±1 standard error across elements.

models (Gemini 2.5 Pro, Claude 3.7 Sonnet, and Deepseek R1 DeepSeek (2025)), all achieving approximately 80%. In general, smaller models (like Llama 4 Maverick Meta (2025), Qwen 2.5 Alibaba (2025), and o3-mini OpenAI (2025b)) performed worse. We also found that newer versions of models generally dominate older versions of the same model, with o3 outperforming o1 OpenAI (2024b) and Gemini 2.5 pro outperforming Gemini 1.5 pro. In general, "reasoning" versions of models seemed to outperform the base versions of the same model. For example, DeepSeek R1 (79.8%) strongly outperformed Deepseek V3 DeepSeek (2024) (72.2%), despite being based on the same 671B parameter architecture. Similarly, Claude 3.7 Sonnet with a thinking budget of 1024 tokens (83.6%) clearly outperformed the base Claude 3.7 Sonnet (80.4%). When compared with total cost of running the benchmark, we see that more expensive models tend to outperform less expensive models, either due to a larger underlying model or using more reasoning tokens in the output.

## 4.2 TRANSFERABILITY OF REASONING SKILLS

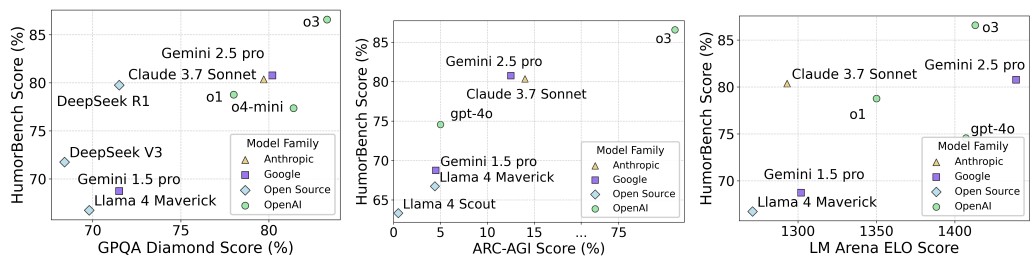

Figure 4: HumorBench performance compared to several benchmarks. We see positive correlation with GPQA, ARC-AGI, and LMArena. In particular, ranking compared to ARC-AGI is nearly identical to that of HumorBench, indicating a strong reasoning component to the HumorBench task.

| Benchmark | Corr. | p-value |
|---|---|---|
| GPQA Diamond | 0.736* | 0.024 |
| ARC-AGI (with o-series) | 0.650 | 0.058 |
| ARC-AGI (w/o o-series) | 0.943** | 0.005 |
| LM Arena ELO | 0.714 | 0.071 |

Table 2: Spearman's rank correlations between HumorBench and other benchmarks. Asterisks indicate significance: $*p < 0.05$, $**p < 0.01$. ARC-AGI correlation shown separately for results with and without o-series models, which were fine-tuned for ARC-AGI

To gauge how well other model skills transfer to humor comprehension, we correlate HumorBench accuracy with three widely used LLM benchmarks: GPQA-Diamond Rein et al. (2023), ARC-AGI Chollet et al. (2025), and LM Arena ELO Chiang et al. (2024). HumorBench scores are positively associated with all three (see table 2). In particular, after removing o-series models (whose scores come from ARC-tuned variants) the correlation with ARC-AGI rises to $\rho = 0.943$ ($p = 0.005$), underscoring the shared skills of the two tasks. The LM Arena correlation ($\rho = 0.714$) is notably lower. Overall, this suggests that LLM progress on STEM domains tranfers to Non-STEM reasoning.

### STEM-ONLY REASONING IMPROVES HUMORBENCH

Our comparison between reasoning models trained on STEM tasks via Reinforcement Learning and their base counterparts yielded particularly revealing results. As illustrated in Figure 6, R1-Zero, which developed reasoning capabilities exclusively through self-play on STEM problems, demonstrated significant improvements over its base V3 model. Remarkably, it performed nearly on par with DeepSeek R1, despite the latter being trained on non-STEM data such as reading comprehension. Similarly, Phi-4 Reasoning Plus exhibited superior performance compared to its base model (Figure 6), although its training was limited to math and coding data (Abdin et al., 2025a). These findings suggest that abstract reasoning capabilities are transferable to humor comprehension, indicating that the reasoning skills required for STEM domains may be fundamentally similar to those needed for understanding humor.

We also note that both R1-Zero and Phi-4 Reasoning Plus include their reasoning traces in their final outputs. Therefore, we evaluated their performances using the length control measure described above to ensure fair comparison across models.

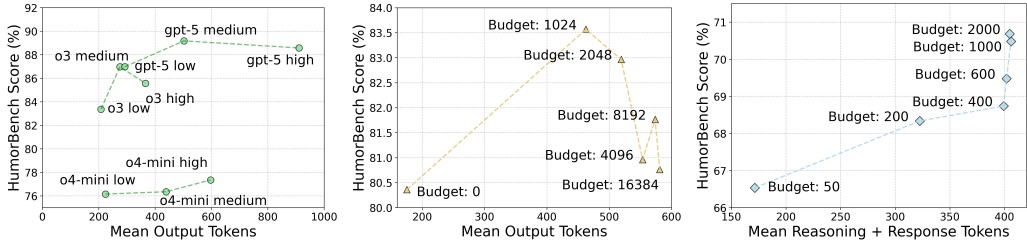

(a) OpenAI reasoning models, varying "reasoning effort"

(b) Claude 3.7 Sonnet performance, varying "thinking budget"

(c) Qwen Plus performance, varying "thinking budget"

Figure 5: HumorBench test-time compute experiments. Note, "mean output tokens" includes both reasoning and final response tokens

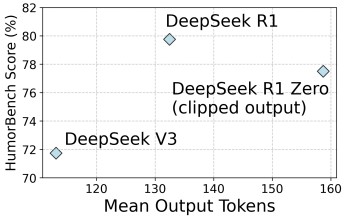
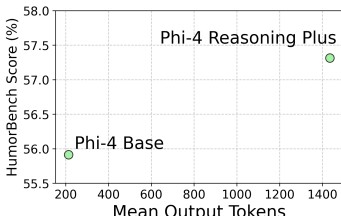

Figure 6: Deepseek R1 Zero and Phi-4 Reasoning Plus, both exclusively reasoning-trained on STEM tasks, outperform their base versions on HumorBench

### 4.3 TEST-TIME SCALING

As seen in figure 5, while including *some* reasoning clearly helped model performance, the effect of continuing to increase test-time compute varied significantly between models. For Qwen plus and the o- series models, increasing the reasoning parameter (reasoning budget and "effort", respectively) generally improved performance. However, for Claude 3.7 Sonnet, increasing the thinking budget beyond the minimum 1024 tokens clearly *hurt* performance.

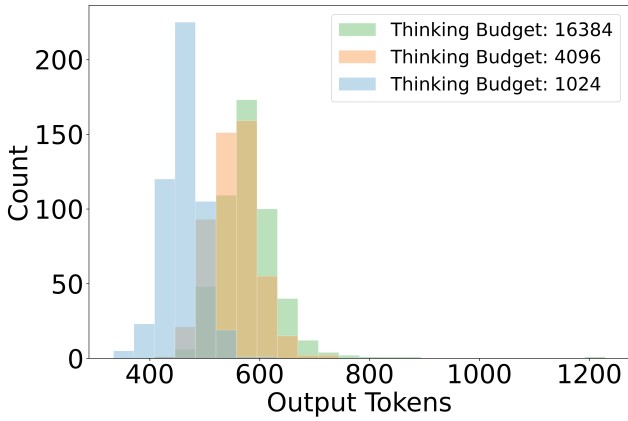

Figure 7: Token usage with different 'thinking budget' parameters on Claude 3.7 sonnet. Most completions (in the hundreds of tokens) were far below their budgets.

A closer look at the reasoning trace lengths highlights that for most captions, the models did not fully exhaust their reasoning budget (see 7). Qwen Plus, for example, rarely used more than 400 tokens, even when budgeted 2000, a trend we saw for all test-time experiments. This suggests the LLMs are providing final answers based on completed thinking traces, which makes the inverse test-time scaling effect more puzzling. A small qualitative trace study for Claude 3.7 Sonnet (Appendix E) points to explicit definition of loaded terms and retrieval of obscure cultural references as one way extra budget helps. While a few studies have looked related problems (Su et al., 2025; Shi et al., 2023; Yang et al., 2025), we defer thorough investigation of our observation to future work.

## 4.4 ANALYSIS OF HUMORBENCH HARD SUBSET

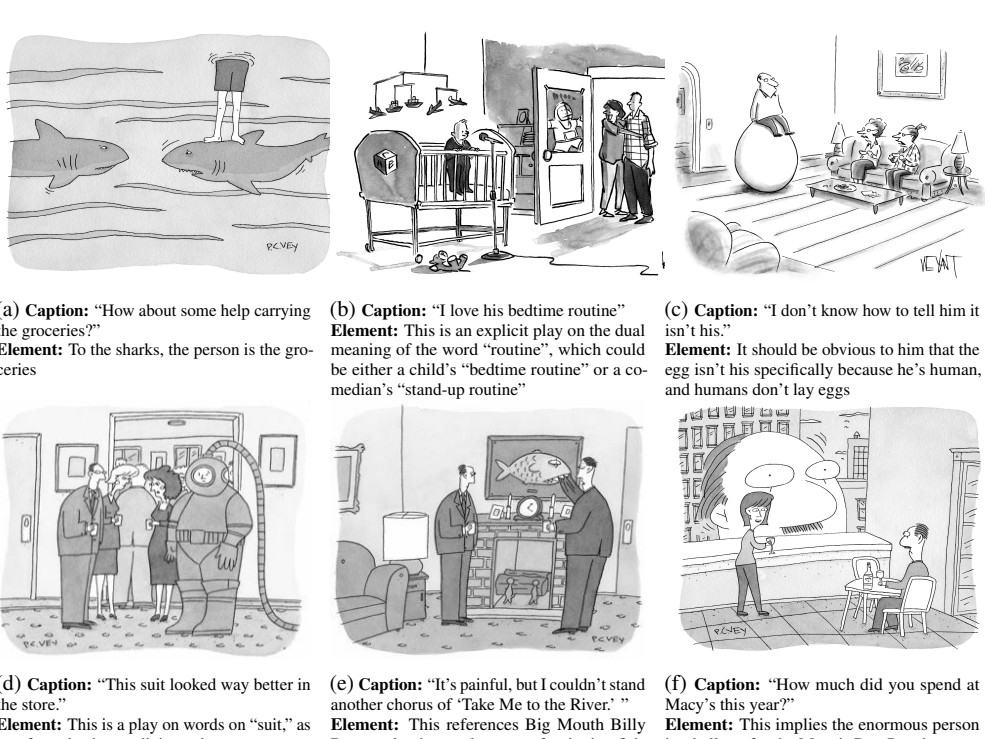

(a) **Caption:** "How about some help carrying the groceries?"
**Element:** To the sharks, the person is the groceries

(b) **Caption:** "I love his bedtime routine"
**Element:** This is an explicit play on the dual meaning of the word "routine", which could be either a child's "bedtime routine" or a comedian's "stand-up routine"

(c) **Caption:** "I don't know how to tell him it isn't his."
**Element:** It should be obvious to him that the egg isn't his specifically because he's human, and humans don't lay eggs

(d) **Caption:** "This suit looked way better in the store."
**Element:** This is a play on words on "suit," as in a formal suit or a diving suit.

(e) **Caption:** "It's painful, but I couldn't stand another chorus of 'Take Me to the River.' "
**Element:** This references Big Mouth Billy Bass, a classic novelty prop of a singing fish, singing "Take Me to the River"

(f) **Caption:** "How much did you spend at Macy's this year?"
**Element:** This implies the enormous person is a balloon for the Macy's Day Parade

Figure 8: Examples of elements in the HUMORBENCH hard subset. See Table 5 for full descriptions.

| Humor Category | Entire Set | Hard Subset | Diff (%) |
|---|---|---|---|
| Wordplay | 24.4% | 19.0% | -5.4 |
| Cultural Reference | 19.0% | 17.7% | -1.3 |
| Toxic or Shocking | 25.7% | 26.6% | +0.9 |

Table 3: Representation of humor categories in the full dataset compared to the hard subset.

While frontier models like o3 demonstrate impressive performance on HumorBench, certain elements are persistently challenging for all models. To better understand the specific types of humor that remain challenging, we conducted a targeted analysis on the 100 unique elements that were most often missed during the benchmarking, which we call HumorBench Hard. Cartoons in this subset range from pass rates of 60% (6 in 10 models get correct) to 0% (No model gets correct). See 3 for examples of the hard subset.

To get a more granular view of the elements that constitute the hard subset, we analyzed three predefined humor categories: *wordplay*, *cultural references*, and *toxic or shocking* humor elements. These categories were annotated by an LLM categorization pipeline built on o3, which individually categorized each element as in or out of each category. Most elements did not fit into these categories, while some fit into multiple. Our analysis examined the relative representation of each category within the hard subset compared to their representation in the overall HumorBench dataset.

We summarize the main findings of this analysis in Table 3. Overall, these relatively minor deviations indicate that humor category alone is not the primary determinant of difficulty for models. Challenging examples likely hinge on subtler factors, such as the implicit conceptual leaps required or the obscurity or references. We observed that wordplay was slightly *under-represented* among the hard subset ($-5.4\%$), suggesting current LLMs handle puns or jokes that rely on linguistic manipulation somewhat better than other elements. *Cultural references* and *toxic or shocking* humor, meanwhile, were essentially evenly represented, indicating that these styles do not disproportionately increase difficulty. These nuanced insights encourage further qualitative investigation into the underlying reasons why particular humor instances remain difficult, even for state-of-the-art LLMs trained explicitly with reasoning capabilities.

Last, in Figure 8, we highlight a few examples that are particularly challenging in the HumorBench hard subset. Detailed descriptions are also provided in Table 5. To our surprise, although several of them are quite intuitive and easy to recognize for humans, LLMs usually struggle on these examples. This represents a fundamental difference in reasoning abilities between humans and LLMs.

## 5 CONCLUSION AND FUTURE WORK

In this work, we introduced **HumorBench**, the first large-scale evaluation that isolates *humor comprehension*, as opposed to subjective funniness, by grading model explanations against concise, expert-annotated *objective elements*. Our experiments with more than a dozen frontier and open-source LLMs revealed that (i) progress on STEM reasoning benchmarks translates strongly to non-STEM humor reasoning, (ii) specialized "reasoning" variants consistently outperform base models even when they were trained only on STEM corpora, and (iii) test-time compute helps, but only up to the point where the relevant background knowledge is actually present in the model. Together, these findings position HumorBench as a sensitive probe of higher-level reasoning that remains comfortably unsolved: the best model still misses over 40% of elements in our hard subset.

Looking forward, there are a number of promising avenues for extending this work. A natural next step is to develop a *multimodal* version of HumorBench by reintroducing the original cartoon images, allowing evaluation of both visual recognition and reasoning and better reflecting the complete humor comprehension task. Alternatively, the use of HumorBench's element rubrics as supervision signals offers an opportunity to explore reinforcement learning. Finally, improving the reliability of LLM-as-judge evaluation remains an important open challenge, particularly for creative tasks.

We hope HumorBench spurs progress on reasoning that bridges the gap between logical deduction and human culture, serving as a springboard for genuinely funny, culturally aware AI systems.

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

## REPRODUCIBILITY STATEMENT

We release all resources needed to reproduce our results in a project repository (the repository will be made public in the non-anonymized version). The repository contains (i) the full benchmark cartoon descriptions and annotations, (ii) the autograder harness that contains the prompts and workflow of the autograder, and (iii) scripts to run the explainer model, run the autograder, and aggregate and analyze results. Detailed instructions are provided in the README

## ETHICS STATEMENT

To respect copyright and enable lawful redistribution, we release only rubric-guided textual descriptions and derived annotations of the cartoons, no original images. Some items may reference sensitive topics common in humor; the benchmark is provided for research use and downstream applications should include appropriate safety review.

## A MAIN BENCHMARK PROMPT

You are a humor expert extraordinaire, judging the New Yorker Cartoon Caption Contest.
Your current task is to help us understand the humor in various submitted captions.
Given a cartoon description and a caption submission, explain (in less than 200
words) *what* the joke is, focusing on the material substance of the joke. STRICTLY
use the format: <explanation>explanation goes here</explanation>
Cartoon description: description Caption: caption

## B AUTOGRADER PROMPT

You will receive: 1. A short cartoon description 2. A winning funny caption 3. A
student's answer 4. A brief "anticipated answer point" that captures the crucial
comedic device or element
Your job is to determine whether the student's answer **explicitly covers** that
"anticipated answer point."
- If the student's answer captures or discusses the key comedic element (even
if the wording is different), **PASS**. - If the student's answer **omits** or
**contradicts** that key comedic element, **FAIL**. - Do not penalize extra details
or expansions. Synonyms or paraphrasing are acceptable if they convey the same
comedic logic. - Be mindful: if the anticipated answer point emphasizes something
specific (e.g. a pun, wordplay, or ironic twist), check that the student's answer
includes it.
At the end of your evaluation, provide exactly two XML tags: 1. <reasoning>Short
explanation of your thought process</reasoning> 2. <judgement>PASS or
FAIL</judgement>
Do not include additional commentary or deviation from this format.
Cartoon description: {description} Caption: {caption} Student's answer:
{explanation} Anticipated answer point:{anticipated point}

## C MODEL PERFORMANCE ON HUMORBENCH HARD

| Model | Accuracy (%) |
|---|---|
| gpt-5 | 61 |
| o3 | 59 |
| Claude 3.7 Sonnet | 54 |
| gemini-2.5-pro-preview-03-25 | 52 |
| deepseek/deepseek-r1-zero | 51 |
| deepseek-ai/DeepSeek-R1 | 51 |
| o1 | 50 |
| o4-mini | 46 |
| Grok 3 | 45 |
| gpt-4o | 42 |
| deepseek-ai/DeepSeek-V3 | 39 |
| meta-llama/Llama-4-Maverick-17B-128E | 35 |
| gemini-1.5-pro | 35 |
| o3-mini | 32 |
| meta-llama/Llama-4-Scout-17B-16E | 29 |
| Qwen/Qwen2.5-72B-Instruct-Turbo | 26 |

Table 4: Accuracy on the **HumorBench-Hard** subset (100 items).

# D  HARD EXAMPLE DETAILS

| Cartoon ID | Description | Caption | Element(s) | Pass Rate (%) |
|---|---|---|---|---|
| CC123065 | Inside a workshop like room, three elves in pointy hats sit at a long table with open laptop computers. The middle elf appears distressed and is speaking, while the two elves on either side look toward him. | "It's from Santa, and it goes way, way beyond jolly." | Frames Santa as a boss making an inappropriate advance on an employee. | 0 |
| NYCC #40 | The cartoon shows a woman in her underwear sitting up in bed, looking forward with a disgruntled expression. A large snow globe with a snowman inside is positioned next to her on the bed. The woman is speaking. | "I think the Manhattan skyline is getting suspicious." | This implies that she is cheating on her partner, a snow globe of the Manhattan skyline, with the snowman. | 17 |
| NYCC #15 | In a restaurant, a man and a woman are sitting down to eat dressed in nice clothing. The man is leaning over the table with his hand on a glass looking at the woman with a soft smile. However, the man is bald and has a cartoonishly large forehead, with the outline of the woman visible on his forehead. The woman, sitting upright, is speaking. | "Well, it's a lovely gesture, but I still think we should start seeing other people." | Implies that the image on his forehead is a tattoo, as getting a tattoo of your significant other is a common practice. | 20 |
| NYCC #669 | A baby leans over the side of a crib toward a microphone on a stand, as if ready to perform. The crib has letter blocks, and a mobile with boats and airplanes hangs above. A teddy bear lies on the floor. In the background, a couple stands in the doorway, looking at the baby with surprise. The woman is speaking with a smile. | "I love his bedtime routine." | Play on the dual meaning of the word "routine": a child's bedtime routine vs. a comedian's stand up routine. | 33 |
| NYCC #665 | Two sharks are facing each other in the ocean. A person, visible only from the waist down, is standing on the back of one of the sharks. The sharks look bewildered; the carrying shark is speaking. | "How about some help carrying the groceries?" | To the sharks, the person is the groceries. | 40 |
| NYCC #687 | A woman holding a wine glass stands on a rooftop in a city, delighted, looking back at a man sitting at a table with a bottle and glass. Behind them, an enormous face peers over the building, resembling the man. The woman is speaking with a smile. | "How much did you spend at Macy's this year?" | Implies the enormous person is a parade balloon for the Macy's Thanksgiving Day Parade. | 40 |
| NYCC #686 | In a living room, a bald man is sitting on a giant egg, looking content. Two older women sipping tea, seated on a couch, are staring at him. The room has a coffee table, lamps, and a framed picture on the wall. One woman is speaking. | "I don't know how to tell him it's not his." | It should be obvious to him that the egg isn't his because humans don't lay eggs. | 27 |
| NYCC #61 | A doctor wearing a head mirror stands behind a desk in a typical office. A giant hand is reaching through the doorway, palm up. The doctor is leaning over to check the enormous hand's pulse. | "I don't know why you're so jolly—your cholesterol is through the roof." | Wordplay: "through the roof" both as extremely elevated levels and literally breaking through the roof. | 20 |

Table 5: Representative examples from the hard subset where a majority of evaluated LLMs failed to identify all required humor elements.

# E  REASONING TRACE CASE STUDIES

To better understand when test-time "thinking" helps, we qualitatively inspected cases where `claude-3-7-sonnet-latest` in thinking mode (budget 2048 tokens) succeeded on an element that the same model in standard mode missed. Prompts, data, and autograder remained identical; only the decoding mode changed.

We highlight two representative anecdotes.

**Anecdote 1: "Faith-based initiative" (NYCC 41.0).**   The cartoon shows a church interior with a fully stocked bar; the caption is *"Finally—a faith-based initiative I can embrace."* The key element is the reference to President George W. Bush's *Faith-Based and Community Initiatives* program.

**Base model.** The standard run defined "faith-based initiative" only in generic terms (e.g., religiously motivated social programs) and never linked it to a specific US policy or administration, so the political reference element was marked as missing.

**Thinking model.** In the thinking trace, the model first writes a short definition of the term as a policy label and then explicitly ties it to the Bush administration before drafting the final explanation:

> "...the phrase 'faith-based initiative' is also a political term, *especially associated with the George W. Bush administration* ..."

This additional retrieval step causes the final explanation to mention the Bush-era program and passes the element.

**Anecdote 2: Big Mouth Billy Bass (NYCC 57.0).**   The cartoon shows a mounted fish on the wall with a man's hand in its mouth; the caption is *"It's painful, but I couldn't stand another chorus of 'Take Me to the River.' "* The key element is that this is a reference to the novelty product *Big Mouth Billy Bass*, a singing fish decoration that plays "Take Me to the River."

**Base model.** The standard run assumes the man is singing the song and interprets the joke as preferring physical pain to bad singing, never mentioning the novelty fish prop, so it fails the element.

**Thinking model.** The thinking trace instead immediately connects the song title to the product:

> "... 'Take Me to the River' is a song that was *famously used for novelty singing fish decorations (like Big Mouth Billy Bass)* ..."

The final explanation then correctly frames the joke around escaping the repeated jingle from the novelty plaque, and the element passes.

These case studies suggest that, at least in some instances, the additional "thinking" budget is used to (i) explicitly define loaded terms and retrieve associated political context, and (ii) connect cues (a song title) to specific cultural artifacts (a novelty product), which can unlock the correct humor interpretation.

## F   WEB SEARCH ABLATION

To assess how access to external knowledge affects humor understanding, we ran an ablation on the GPT-5 model family. For each of three sizes (Nano, Mini, and full) we compared a *base* configuration (no tools) to a *search* configuration in which the model could call a web search tool before producing its explanation. Prompts, HumorBench items, and the GPT-4o autograder were identical across conditions, and evaluated on the full benchmark as well as the hard subset.

| Model | Overall, Base (%) | Overall, Search (%) | Hard, Base (%) | Hard, Search (%) |
|---|---|---|---|---|
| GPT-5 Nano | 69.1 | 67.9 | 34.0 | 31.0 |
| GPT-5 Mini | 77.8 | 80.2 | 41.0 | 47.0 |
| GPT-5 | 87.2 | 89.4 | 56.0 | 63.0 |

Table 6: HumorBench element-level accuracy (%) for GPT-5 models with and without web search, on the full benchmark (Overall) and the 100 hardest items (Hard-100).

Overall, web search slightly hurts GPT-5 Nano, but consistently improves GPT-5 Mini and GPT-5. This pattern suggests that external tools are most useful when the base model is already strong enough to recognize when and how to use retrieved information.

