# OpenReview forum: "Which LLMs Get the Joke? Probing Non-STEM Reasoning Abilities with HumorBench"
_ICLR.cc/2026/Conference — ICLR 2026 Conference Desk Rejected Submission_

### Official Review · Reviewer_ATma · 2025-10-30

**Soundness:** 3
**Presentation:** 3
**Contribution:** 3
**Rating:** 4
**Confidence:** 4

**Summary:**

This paper introduces HumorBench, a benchmark designed to evaluate LLM ability to reason about and explain sophisticated humor in cartoon-caption settings. It highlights the limitations of existing humor datasets, which often conflate subjective funniness with objective joke comprehension, and instead proposes a framework focused on objective, verifiable joke elements. To address this gap, the authors curate about 300 cartoon-caption pairs from the New Yorker Caption Contest and Cartoonstock, annotate each with core joke elements, and build an autograder to assess whether LLM explanations capture those elements. Through experiments with frontier models, the study shows (1) strong transfer of reasoning skills from STEM benchmarks to humor comprehension, (2) surprising competence of models trained only on STEM reasoning tasks, and (3) mixed effects of test-time scaling.

**Strengths:**

1.	The paper identifies a real gap in LLM evaluation: existing benchmarks overemphasize STEM reasoning, while humor demands cultural, linguistic, and inferential reasoning.

2. The dataset is constructed with rigorous curation, including expert validation and removal of inconsistent annotations, contributing to dataset quality and reliability.

3.	It provides systematic evaluation, tests across many models, compares base vs. reasoning optimized variants, and analyzes correlations with other benchmarks.

**Weaknesses:**

1.	The size of the dataset may be limited. Although curated carefully, about 300 cartoon-caption pairs (499 elements) is relatively small compared to other reasoning benchmarks, potentially limiting generalization.
2.	Need further justification for the dataset diversity, especially w.r.t. cultural background and humor type. Considering the original sources are largely Western (New Yorker, Cartoonstock), and the dataset may not test models across diverse humor traditions.
3.	The validation shows a leniency bias (higher false positive rate), meaning scores are optimistic upper bounds. The reliance on LLM-as-judge could compromise fairness and transparency.
4.	While the benchmark shows transfer from STEM reasoning, the interpretation sometimes overstates the connection without fully probing why abstract reasoning transfers across domains.
5.	The explanations of why models fail certain jokes remain somewhat descriptive rather than deeply diagnostic.

**Questions:**

1.	The benchmark currently relies on textual descriptions of cartoons rather than the original images. How might removing the visual modality bias the evaluation? Do you plan to extend HumorBench into a multimodal setting?
2.	How do you think HumorBench generalizes across different cultures and humor traditions? Do you envision multilingual or cross-cultural extensions?
3.	Do you have hypotheses about why this “inverse scaling” occurs, and what it implies about LLM reasoning dynamics?
4.	Would you incorporate both objective and subjective dimensions in future benchmarks?

---

> ### Author Response · Authors · 2025-11-22
>
> Dear reviewer,
>
> We appreciate your review, and address your points below:
>
> Benchmark size:
> Correct annotations in this domain are difficult to produce, requiring deep familiarity with the format and verification by our expert former New Yorker Chief Cartoon Editor. As such, we emphasized data quality over scale. This is in line with several widely adopted industry benchmarks, such as SWEBench-verified and GPQA, which have 500 and 488 annotation points, respectively.
>
> Cultural background and potential bias:
> Indeed, the NYCC caters to a distinct, Western cultural background. While caption submissions are received from around the world, the ultimate finalists are chosen by the New Yorker editorial team, who represent a particular humor taste. We have updated the wording in Section 1.1 (lines 105-107) to narrow claims about how broadly this applies. We agree that follow-up work probing other cultural backgrounds would add depth to the claim, but such a dataset is not readily available.
>
> False Positive Bias:
> To ensure fair evaluation, we detailed our separate analysis of the autograder itself in Section 3.5, finding that it agrees with human judgments 92% of the time, which is higher than most LLM-as-a-judge accuracies. OpenAI’s PaperBench, for example, has an autograder accuracy of ~80%, and G-Eval (EMNLP 2023) has an autograder accuracy of ~85%.
> While our autograders are generally more lenient (indicated by higher false positive rates), their leniency is consistent across both autograder and explainer models, indicating a valid source of signal from the benchmark.
>
> Deeper analysis of the reasoning improvement claim:
> We have done deeper inspection on the reasoning traces, which we will add to the appendix. The most common elements that reasoning improved upon were deep cultural references, particularly when the model enumerates all the possible ways the caption might work, retrospectively identifying these as the “real” joke. We believe this demonstrates an interesting search/retrieval pattern surfaced by the reasoning.
>
> Analysis why models fail certain jokes:
> The reason certain captions are difficult is difficult to pin down; We’ve added Appendix F, where we inspect anecdotes of instances where a reasoning model gets it right, but a non-reasoning model misses an element, along with the Section of the reasoning trace where the model “gets it.” We found that subtler cultural references sometimes take more test-time compute to stumble upon, which we think of as being harder because they are deeper in the model’s search space. However, other aspects such as the wording of the caption (how explicitly it calls out a certain reference or wordplay, for example) also influence the difficulty, making it difficult to pin down.
>
> Effect of removing images:
> While the information carried in the image is often important to understanding a caption, it is worth emphasizing that all the captions are submitted to the same cartoon. Thus, the way the winning caption’s humor separates it from the other submissions is typically separate from the style of the image. We annotated the image descriptions with the winning captions in mind, so that relevant information was never left out. Thus, we believe we are not biasing the evaluation by removing the image.
>
> We removed the images to respect the IP of the artists, and not redistribute the original images without purchasing licenses for them, which would far exceed the budget of our project. However, it may be reasonable for an industry company to do. We invite this, and provide source links in the dataset to make it easy to extend to an image-included setting.
>
> Inverse test-time compute scaling observation:
> Because this was not observed across all models, pinning down why more tokens sometimes hurts performance is difficult, as all models have different versions of posttraining and reasoning length control. One relevant point (see figure 7) is that after a certain threshold thinking budget, Claude 3.7 Sonnet tends not to increase its actual output length. We believe that by predefining a reasoning budget, you add bias to the model’s reasoning process, asking it to do more reasoning than is necessary.
>
> Would we include subjective elements in the future?
> To maintain the clean signal from the dataset, we were very intentional about removing subjective elements from the annotations, which may introduce noise to the evaluation. Subjective elements could cause reasonable interpretations of the caption to be graded as wrong. Thus, we do not plan on including subjective elements in future work, without further agreements in psychology literature surrounding the underlying mechanisms of humor.
>
> Thank you again for your detailed and thoughtful feedback, and we hope that we’ve addressed your questions and concerns.

---

### Official Review · Reviewer_eZVy · 2025-10-30

**Soundness:** 2
**Presentation:** 3
**Contribution:** 2
**Rating:** 4
**Confidence:** 4

**Summary:**

This paper introduces HumorBench, a new benchmark designed for evaluating humor understanding in models. HumorBench comprises 300 unique cartoon–caption pairs collected from the New Yorker Caption Contest and Cartoonstock.com, each annotated with human-defined rubrics. Unlike prior humor understanding benchmarks, HumorBench decomposes humor into several objective elements and assesses a model’s generated explanations against these elements using an automatic grading system. This approach aims to measure objective comprehension of humor rather than subjective interpretations. The authors evaluate several recent multimodal large language models (MLLMs) on HumorBench and present analyses that reveal their relative performance and limitations.

**Strengths:**

- The introduction of a rubric-based evaluation framework for humor understanding is novel and well-motivated.
- The proposed HumorBench benchmark has the potential to offer valuable insights and guide future research in multimodal humor understanding.
- The experimental design and analyses are comprehensive, effectively highlighting the limitations and challenges faced by current MLLMs in this domain.

**Weaknesses:**

- The benchmark size is relatively small, containing only about 300 cartoons drawn from a limited range of sources (mainly The New Yorker Caption Contest and Cartoonstock.com). This restricted scope may limit the dataset’s generalizability and future applicability;
- While the authors argue that using humor elements as rubrics enables objective humor understanding, this claim raises concerns. The definition and categorization of these humor elements may be subjective and culturally biased, as different annotators or readers could conceptualize these humor elements differently. The paper would benefit from a clearer justification of how these elements were defined and whether they adequately capture the diversity of humor mechanisms. Moreover, important annotation details are missing—including the number and backgrounds of annotators (particularly their cultural contexts), and whether inter-annotator agreement was measured to ensure annotation reliability;
- The paper reports a correlation between STEM reasoning ability and humor understanding, but this interpretation may be confounded. The observed correlation could simply reflect that stronger MLLMs perform better across both STEM and Humor reasoning domains, rather than implying a causal link. Without a controlled experiment, for instance, fine-tuning the same base model on STEM data and comparing its humor understanding performance, the claim that enhancing STEM reasoning improves humor comprehension remains unsubstantiated;
- Regarding Ethics Statement: Including the background of annotators and how you ensure the annotations are not biased.

**Questions:**

See Weaknesses

---

> ### Author Response · Authors · 2025-11-22
>
> Dear reviewer,
> We appreciate your review, and address each concern below:
>
> Benchmark size:
> Correct annotations in this domain are difficult to produce, requiring deep familiarity with the format and verification by our expert former New Yorker Chief Cartoon Editor. As such, we emphasized data quality on all 499 element annotations. This is in line with several widely adopted industry benchmarks, such as SWEBench-verified and GPQA, which have 500 and 488 annotation points, respectively.
>
> Annotation details and cultural background:
> Thank you for pointing out the potential for cultural bias. We agree that only testing New Yorker readership culture in the annotations is limiting, and performance on cartoons from another humor culture may result in different relative ordering of models-- we’ve explicitly added this to lines 105-107. To ensure this benchmark is a fair evaluation, we emphasized in the annotation process (see Section 3) that we only test for verifiable aspects of a joke (e.g. concrete wordplays). This could be though of as probing for the specific world knowledge that makes the caption make sense. To further improve annotation accuracy, we had a former New Yorker Chief Cartoon Editor review and verify annotations. There were two annotators, plus the Editor who contributed to the dataset. The annotators were both longtime participants in the contest, which we will further specify in the paper.
>
> STEM reasoning claim:
> We did run a controlled experiment of base models (Deepseek V3, Phi-4) and their STEM-only-trained reasoning counterparts (Deepseek R1, Deepseek R1 Zero, and Phi-4 Reasoning Plus), finding that the reasoning training improved performance in all cases. This is detailed in the second half of section 4.2.
>
> To bolster the claim that there has been an abstract reasoning skill learned that improves performance, we performed deeper inspection on the reasoning traces, which we have now added to the appendix. The most common elements that reasoning improved upon were deep cultural references, particularly when the model enumerates all the possible ways the caption might work, retrospectively identifying these as the “real” joke. We believe this demonstrates an interesting search/retrieval pattern surfaced by the reasoning.
>
>
> Thank you again for these comments, and we think these tweaks improve the paper.

---

### Official Review · Reviewer_Ruae · 2025-10-31

**Soundness:** 2
**Presentation:** 3
**Contribution:** 2
**Rating:** 4
**Confidence:** 3

**Summary:**

This paper presents HumorBench, a benchmark designed to evaluate the ability of large language models (LLMs) to reason about and explain sophisticated humour in cartoon captions. It includes approximately 300 unique cartoon-caption pairs from the New Yorker Caption Contest and Cartoonstock.com, with expert-annotated evaluation rubrics identifying essential joke elements. The authors also conducted a rich experiment by benchmarking current SOTA models and provided several insights for future work.

**Strengths:**

1. HumorBench introduces a new non-STEM reasoning task that isolates objective humour understanding, avoiding confounding subjective funniness and providing a valuable probe of high-level reasoning.

2. Through extensive experiments on frontier LLMs, the study reveals clear transfer from STEM reasoning to humour comprehension and mixed effects of test-time scaling, demonstrating both the benchmark’s sensitivity and the cross-domain generality of reasoning abilities.

**Weaknesses:**

1. Besides identifying individual elements, the evaluation should also consider the interactions or causal relations among them. Humour understanding depends on how well the model connects these elements, not just mentions them. Element-based evaluation can serve as a supplement. In fact, using an LLM as a judge to compare the response with the golden label can already effectively reflect joke understanding.

2. This conclusion is reasonable, but the paper does not explain why reasoning models improve humour understanding. The causes should be analysed further: Is the gain due to longer thinking tokens, potential training data leakage, or emergent self-search/retrieval behaviour of the model?

3. Although the text-only setup makes the task more controllable and annotations clearer, it may oversimplify the problem. Under this setting, GPT-4o already achieves around 92% accuracy, suggesting that the task ceiling may be too low to differentiate between stronger models.

4. Models with access to external tools (e.g., web search) may perform better in humour reasoning. Since the paper mentions that many hard cases involve hidden concept jumps or obscure references, evaluating models equipped with such tools would provide more meaningful insights.

**Questions:**

See above.

---

> ### Author Response · Authors · 2025-11-22
>
> Dear reviewer,
>
> We appreciate your review, and address each concern below.
>
> Interaction between elements:
> While interactions between elements may constitute parts of why a caption is funny, pinning down their interaction in an objective way is typically not possible. For example, one explanation of an interaction could be that the two elements both share a similar emotional valence, enhancing the overall effect because it is a stronger emotion. An equally valid interpretation could explain the effect as being sarcastic because of being over-the-top. When different but reasonable interpretations of an element are present, fairly judging an explanation becomes subjective and thus noisy. For this reason, we stick with annotations that are single, verifiable components of the joke.
>
> Deeper analysis on why reasoning improves performance:
> We have done deeper inspection on the reasoning traces, which we have added to the appendix. The most common elements that reasoning improved upon were deep cultural references, particularly when the model enumerates all the possible ways the caption might work, retrospectively identifying these as the “real” joke. One illustrative example:
> In contest 41, the caption “Finally, a faith-based initiative I can embrace” is a nod to the George W. Bush presidency, which made “faith based initiatives” a recognizable phrase. One of the annotations of this cartoon is this reference to George W. Bush. While the base model misses the George Bush reference, the reasoning model gets it. In its chain of thought, we can see that it enumerates possible elements of the joke internally, only mentioning George Bush on the fifth bullet point. Thus, through the reasoning process, it searched and found the relevant connection. We believe this demonstrates a search/retrieval pattern surfaced by the reasoning.
>
> Task ceiling:
> To clarify, 92% was gpt-4o’s score as the autograder, in agreement with human judgements. This is distinct from the benchmark performance, where gpt-4o achieves 74.5%.
>
> While current SOTA models do perform well on this benchmark, it is worth noting that recently released models (GPT-5, Grok-4, Gemini 3) continue to close the gap, indicating that there is still signal in the remaining ~10-20% of incorrect answers.
>
> Web search access:
> We did new runs with web search, and found that indeed, web search improves performance of gpt-5 at mini- and regular- sizes, but not nano-. In fact, GPT-5 + web search achieved the best performance so far, of ~91%. Note, however, that on the hard subset, GPT-5 + search still only achieves 63%, indicating that the benchmark is not yet saturated, and that there is still room to improve. These results have been added to the paper as Appendix F.
>
> Thank you again for these constructive suggestions; we believe these new analyses (reasoning traces and web-search ablations) significantly strengthen the paper.

---

### Official Review · Reviewer_rg5y · 2025-11-01

**Soundness:** 3
**Presentation:** 3
**Contribution:** 2
**Rating:** 6
**Confidence:** 3

**Summary:**

This paper constructs a new benchmark based on the New Yorker Cartoon Contest and CartoonStock datasets, through re-annotation and the removal of images. Unlike previous datasets, HumorBench relies solely on objective elements that aid in humor understanding and excludes images, making it a cleaner benchmark for evaluating humor comprehension.

**Strengths:**

- During the annotation of the dataset, elements that could subjectively influence humor understanding were deliberately removed.
- The authors avoid using multiple-choice and ranking formats, as these can limit the model’s reasonable divergence in humor understanding. Additionally, fixed options may inadvertently hint at the actual punchline, making it unclear whether the model is reasoning or simply guessing. This is a reasonable and effective improvement.
- The authors designed a multi-round data refinement method that uses two large language models to alternately evaluate the data.

**Weaknesses:**

- Some steps in the dataset construction process are not described in detail. (See Questions.)
- The absence of images in the dataset may weaken its impact, as the main limitation of current multimodal large models lies in their inability to effectively understand humor from images.

**Questions:**

- What is the criterion you use to distinguish between subjective and objective elements?
- Why did you choose “whether it contains a joke element” as the evaluation criterion for the autogravder, rather than evaluating the complete output?
- Do you have plans to extend the dataset to include a version with images?

---

> ### Author Response · Authors · 2025-11-22
>
> Dear reviewer,
>
> We appreciate your review, and address your questions below:
>
> Subjective/objective distinction:
> We’ve clarified in section 3.3 how our distinction between objective and subjective elements, thank you for pointing this out. Primarily, we draw the line at verifiability. An objective element is a short, necessary fact or relation that a correct explanation must include and that can be checked from the caption/description plus widely shared background knowledge (e.g., a named cultural referent, a concrete wordplay relation). Subjective aspects would make arguments about how humor itself works, which is still largely disagreed upon in psychology. This would add noise to the benchmark, and was thus carefully avoided.
>
> Why grade per- element?
> Cartoon captions often hinge on multiple mechanisms; a model may correctly identify one mechanism (e.g., a cultural reference) while missing another (e.g., a wordplay). Element‑level checks allow principled partial credit and make open‑ended explanations reproducibly gradable (see sections 3.1, 3.3), effectively serving as a rubric on a per-cartoon basis. This is a commonly used strategy, employed in benchmarks such as OpenAI’s paperbench. Empirically, when we also aggregate to a per‑caption “all‑elements‑required” score, the relative ordering of models is essentially unchanged.
>
> Why exclude images?
> To respect the IP of the artists, we cannot redistribute the original images without purchasing licenses for both Cartoonstock and NYCC cartoons, which far exceed the budget of the project. However, it may be reasonable for an industry company to run. We invite this, and provide source links to make it easy to extend to an image-included setting.
>
> Thank you again for your thoughtful questions and suggestions; we believe the resulting clarifications and analyses have strengthened the paper

---

### Note · Program_Chairs · 2026-01-17
**Submission Desk Rejected by Program Chairs**

The following references in this submission do not refer to real documents and/or have major errors in bibliographic information:

 Matteo Starace, Adam Fisch, Nora Kassner, Jason Wei, Johannes Welbl, Marjan Ghazvininejad, Sebastian Riedel, and Jack Merullo. Paperbench: Benchmarking large language models for scientific paper understanding and knowledge production. arXiv preprint arXiv:2501.12077, 2025.
Liangyi Yang, Yihuai Wang, Xinyan Du, Zhekai Zhang, Haotian Shi, Zhengying Zhu, et al. Mathglm: Towards generalizable mathematical reasoning via language modeling. arXiv preprint arXiv:2401.14273, 2024.